# IL-17 Promotes Nitric Oxide Production in Non-Small-Cell Lung Cancer

**DOI:** 10.3390/jcm10194572

**Published:** 2021-10-01

**Authors:** Stefania Nicola, Irene Ridolfi, Giovanni Rolla, Pierluigi Filosso, Roberto Giobbe, Monica Boita, Beatrice Culla, Caterina Bucca, Paolo Solidoro, Luisa Brussino

**Affiliations:** 1Allergy and Clinical Immunology Unit, Department of Medical Sciences, University of Torino & Mauriziano Hospital, 10128 Turin, Italy; irene.ridolfi@edu.unito.it (I.R.); grolla@mauriziano.it (G.R.); monica.boita@libero.it (M.B.); bea_culla@hotmail.com (B.C.); caterina.bucca@unito.it (C.B.); luisa.brussino@unito.it (L.B.); 2Department of Thoracic Surgery, University of Torino, Ospedale Molinette, Via Genova 3, 10128 Turin, Italy; pierluigi.filosso@unito.it (P.F.); roberto.giobbe@gmail.com (R.G.); 3S.C. Pneumologia U, Azienda Ospedaliero Universitaria Città della Salute e della Scienza, 10128 Turin, Italy; 4Department of Medical Sciences, University of Turin, 10128 Turin, Italy; paolo.solidoro@unito.it

**Keywords:** Th-17 cells, IL-17, IL-23, NSCLC, non-small-cell lung cancer, FeNO, JawNO, CalvNO, lung malignancy, EBC, exhaled breath condensate, VEGF

## Abstract

Introduction: Lung cancer is the second most frequent malignancy worldwide, but its aetiology is still unclear. Inflammatory cytokines and Th cells, including Th17, are now emerging as being involved in NSCLC pathways, thus postulating a role of IL-17 in tumour angiogenesis by stimulating the vascular endothelial growth factor and the release of nitric oxide. Despite the fact that many biomarkers are used for chest malignancy diagnosis, data on FeNO levels and inflammatory cytokines in NSCLC are still few. Our study aimed to evaluate the relationship between pulmonary nitric oxide production and VEGF and Th17-related cytokines in the EBC of patients affected by early-stage NSCLC. Methods: FeNO measurement and lung function tests were performed in both patients affected by NCSLC and controls; EBC samples were also taken, and Th1 (IL-1, IL-6, IL-12, IFN-g, TNF-a), Th17 (IL-17, IL-23) and Th2 (IL-4, IL-5, IL-13) related cytokines were measured. Results: Th1 and Th17-related cytokines in EBC, except for IFN-gamma and TNF-alpha, were significantly higher in patients than in healthy controls, whereas no differences were seen for Th2-related cytokines. FeNO at the flow rate of 50 mL/s, JawNO and CalvNO levels were significantly higher in patients affected by NSCLC compared to controls. Significant correlations were found between FeNO 50 mL/s and IL-17, IL-1 and VEGF. JawNO levels positively correlated with IL-6, IL-17 and VEGF. No correlations were found between FeNO and Th2-related cytokines. Conclusion: This is the first report assessing a relationship between FeNO levels and Th17-related cytokines in the EBC of patients affected by early-stage NSCLC. IL-17, which could promote angiogenesis through the VEGF pathway, might be indirectly responsible for the increased lung production of NO in patients with NSCLC.

## 1. Introduction

Lung cancer is the second most frequent malignancy worldwide [1], and it is responsible for about 25% of all cancer deaths [2]. Among many types of lung malignancies, non-small-cell lung cancer (NSCLC) accounts for over 80% of cases, and it is mainly associated with smoking habits [3].

Although much effort has been made, the aetiology of lung cancer is still not completely understood. However, a synergy between environmental and genetic factors, including gene polymorphisms and cytokine expression, was postulated as having a pivotal role in susceptibility to lung malignancy [4,5].

Over the years, research has focused on inflammatory pathways and cytokine networks, which might promote lung cancer [6]. Besides the well-known role of nitric oxide (NO) in type 2 (T2) airway inflammation and the role of T-helper (Th)1 and Th2 cells in lung cancer pathogenesis [7], Th17 cells are now emerging as actively involved in NSCLC pathways [8,9].

Th17 cells have been shown to be highly expressed in many inflammatory and autoimmune disorders, including systemic sclerosis [10,11,12], psoriatic arthritis [13] and asthma [14], thus becoming therapeutic targets in many diseases.

Recently, interleukin (IL)-17A, which is a proinflammatory cytokine secreted by Th17 lymphocytes, has been found overexpressed in many malignancies [15,16,17,18], including NSCLC. In addition, IL-17A levels have been associated with higher tumour vessel density and more advanced cancer stage, thus raising the hypothesis that IL-17A could be involved in malignancy progression [15].

As a result, many authors [16,17,18,19] have hypothesised that IL-17 might promote tumour angiogenesis by stimulating the vascular endothelial growth factor (VEGF) [20], which, in turn, may increase microvascular permeability and the release of endothelial-derived nitric oxide (NO) [21,22,23].

We also demonstrated that IL-17 and VEGF were significantly higher in patients with NSCLC than controls, and their levels significantly correlated with tumour size [19].

Fractional exhaled nitric oxide (FeNO) is a reliable way to measure NO lung production, and is currently used not only in the evaluation of T2 airway inflammation [24], but also as a biomarker of steroid responsiveness in asthma and chronic obstructive pulmonary disease (COPD) [25]. Moreover, NO is involved in lung function impairment and pulmonary hypertension in systemic sclerosis [26,27].

Breath analysis is becoming more and more used in the diagnosis and monitoring of inflammatory lung diseases, as it is a non-invasive technique. Through a tidal breath, exhaled breath condensate (EBC) gives the opportunity to obtain a representative sample of lower respiratory tract fluid, in which many cytokines and inflammatory markers could be searched for [28,29,30].

To our knowledge, even if many biomarkers have been recently reported for early detection and follow-up of chest malignancies [31], data on FeNO levels and inflammatory cytokines in NSCLC are still few [32].

Our study aimed to evaluate the relationship between pulmonary nitric oxide production, VEGF and Th17-related cytokines, measured in the EBC of patients affected by early-stage NSCLC.

## 2. Materials and Methods

### 2.1. Patients

Fifteen adult patients affected by NCSLC and scheduled for curative lung resection at the Thoracic Surgery Unit of the University of Turin, between January and June 2011, were enrolled in the present study. Lung cancer stages and progression were categorised according to the TNM Classification of Malignant Tumours [33].

Thirty healthy subjects were also enrolled as a control group. They had no history of cancer and were matched for gender, age and smoking habits with the patients’ cohort.

The study was approved by the Institutional Review Board for human studies (n. 0039 653) and it was performed according to the Helsinki Declaration.

All patients provided their written informed consent before they were enrolled.

### 2.2. Inclusion and Exclusion Criteria

All enrolled patients were former smokers, and they all showed a peripheral tumour mass. Only patients affected by early-stage NSCLC [33], for whom lung resection was completely curative, were enrolled in the study.

Patients with COPD, asthma, interstitial lung disease, autoimmune disorders, and patients who received neoadjuvant chemotherapy or oral/inhaled corticosteroids were excluded from the study. In addition, upper or lower airway infections within 8 weeks before the screening visit were also considered an exclusion criterion.

### 2.3. Exhaled Breath Condensate Collection

In both patients and controls, EBC collection had been obtained in the same lab room.

Exhaled breath condensate was acquired through an iced disposable R-Tube™ (Respiratory Research Inc., Charlottesville, VA, USA) after mouth rinsing with water. All enrolled subjects tidally breathed through a mouthpiece linked to the R-Tube™ for about 10 min. They also wore a nose clip to avoid nose breathing. EBC collection was temporarily stopped and resumed in case of cough or excessive saliva production. At least 2 mL of EBC was collected from each patient and control. All EBC samples were stored at −80 °C in polypropylene tubes [34].

All EBC samples were tested for amylase activity to exclude saliva contamination (alpha-Amylase ESP1491300 kit; detection limit 0.05 mmol/L; Boehringer Mannheim, Germany) and protein amount was evaluated through a Quantipro BCA assay kit (Sigma Aldrich, Sydney, Australia).

### 2.4. Cytokine Assays

EBC cytokines were analyzed using a multiplex immunoassay on Bioplex 100 instrument (Bio-Rad Laboratories Inc., Hercules, CA, USA). An x-MAP technology (Luminex Corp, Austin, TX, USA) was used for measurement of both cytokines and VEGF, and Bioplex Manager 4.1 software (Bio-Rad Laboratories) was used for data analysis [35,36]. EBC samples of patients and controls were accurately distributed and analyzed on the same assay plate, according to the manufacturer’s recommendations [37,38]. All concentrations are expressed in pg/mL, if not else specified. Data analysis was performed only for cytokines, whose concentration could be detected in more than half of the samples.

### 2.5. Pulmonary Function Tests

Lung function tests were performed using Baires System (Biomedin, Padua, Italy) in both patients and controls, according to the American Thoracic Society (ATS) guidelines [39]. At least three different flow–volume curves with vital capacity (VC) variations within 5% were recorded. The values of VC, Forced Expiratory Volume in the 1st second (FEV1), and FEV1/VC% ratio were also stored. Data recorded from lung function tests were then matched with the predicted values suggested by Quanjer and Cotes [40,41].

Before starting lung function tests, the ID number, name, sex, age, weight and height were recorded.

### 2.6. FeNO

FeNO measurement was performed in all patients and controls before the lung function test, following the ERS/ATS guidelines [42,43]. NO measurement was performed using a chemiluminescent analyser (Sievers NOA 280; Sievers, Boulder, CO, USA), which was previously calibrated with a certified NO calibration gas mixture. The standard expiratory flow rate for NO evaluation was 50 mL/s.

After a maximal inhalation of “NO-free” air (i.e., <1 ppb) through an acid gas filter (NO, <5 parts per billion (ppb)) (AFL 1410; Sievers), subjects exhaled into a mouthpiece connected to a one-way valve, which contained two sampling ports. The first drove nitric oxide directly into the analyser, while the second port measured the exhalation mouth pressure.

By varying the flow resistance of the expiratory circuit, the NO flow rate could have been changed, and the subject was required to breathe out with a greater force.

Resistance was changed between 11 and 16 cm H_2_O, so that nitric oxide was sampled at flow rates of 50, 150 and 200 mL/s. FeNO values below 25 ppb were considered normal [42,43].

In addition, using the two-compartment model described by Tsoukias and George [44], we also assessed the NO output of the bronchial airways (in pL/s) (the intercept JawNO) and the alveolar NO concentration (in ppb or nL/L; the slope CalvNO).

### 2.7. Statistical Analysis

Statistical analysis was performed using a commercially statistical package (STATA 10 s).

All data were firstly screened for normality of the variables’ distribution (Kolmogorov–Smirnov, Shapiro–Wilk and D’Agostino’s K-squared).

Statistical comparison among cytokines was made using either the Mann–Whitney U test or Student’s t-test, according to the normal distribution of variables, and the results were considered as statistically significant only if *p* values were below 0.05.

Correlations between cytokines and lung function tests were analyzed by regression analysis. Spearman’s rank correlation coefficient or Pearson’s correlation coefficient were performed in accordance with the normality distribution of data [45,46].

## 3. Results

Fifteen consecutive patients (12 males and 3 females) affected by early-stage NSCLC were enrolled in the study. A group of 30 healthy people (21 males, 9 females), matched for age, gender and smoking history, was also enrolled as a control group.

The mean age was 63.8 (range 39–82) years in the NCSCL group and 60.1 (range 41–79) years in the control group.

Eight patients had an IA cancer stage, four patients were affected by an IB clinical stage and three of them had an IIA lung cancer stage [33] (Table 1).

All cytokines were detectable in more than half of the samples, and no significant correlations among them were found.

### 3.1. Comparison between Th1, Th2 and Th17-Related Cytokines in Patients and Healthy Controls

Th1 and Th17-related cytokines in EBC, except for IFN-gamma and TNF-alpha, were significantly higher in patients than in healthy controls (Table 2).

On the other hand, no significant differences were found for Th2-related cytokines between patients and controls (Table 2).

### 3.2. Comparison between VEGF in Patients and Healthy Controls

VEGF concentrations in EBC were significantly higher in patients affected by NSCLC than in healthy controls (Table 2).

### 3.3. Lung Function Tests and FeNO Levels in Patients and Controls

No differences were found in FEV1, FVC and FEV1/FVC between patients and controls (Table 3).

FeNO at the flow rate of 50 mL/s, JawNO and CalvNO levels were significantly higher in patients affected by NSCLC compared to controls, whereas no differences were found in levels of FENO at 150 mL/s between the two groups (Table 3).

### 3.4. Correlations between Cytokines and FeNO in NSCLC Patients

Significant correlations were found between FeNO 50 mL/s and IL-17 (r = 0.467, *p* = 0.049), IL-1 (r = 0.841, *p* = 0.036) and VEGF (r = 0.754, *p* = 0.044). JawNO levels positively correlated with IL-6 (r = 0.502, *p* = 0.042), IL-17 (r = 0.796, *p* < 0.001) (Figure 1) and VEGF (r = 0.761, *p* = 0.001) (Figure 2) (Table 4). No correlations were found between FeNO 50 mL/s as well FeNO 150 mL/s and Th2-related cytokines (data not shown).

## 4. Discussion

This is the first report assessing a relationship between FeNO levels and Th17-related cytokines measured in EBC in patients affected by early-stage NSCLC.

Our results show that most Th1 and Th17-related cytokines, including IL-17, were significantly higher in the EBC of patients affected by NSCLC compared to healthy controls. This finding perfectly correlates with the well-known lung inflammation degree, which is a feature of lung cancer [7,8].

In a previously published paper [19], we already showed the relationship between IL-17 in EBC and tumour size in NSCLC, thus postulating an involvement of Th17 cells in the progression of the malignancy.

Here, not only did we confirm the increased production of IL-17 in patients with NSCLC, but we also found correlations between IL-17 and FeNO levels, which is an absolute novelty of our study.

Our results show, in fact, that FeNO levels, which were significantly higher in patients compared with healthy controls, positively correlated with IL-17 production, thus confirming the inflammatory process underlying the malignancy.

However, even though FeNO is a well-known T2 inflammation marker, and its production is strictly dependent on the activation of the IL-4/IL-13 axis [47], our results showed no differences between Th2-related cytokines between patients and controls. In addition, our patients were not affected by any inflammatory or autoimmune lung disorder, thus raising the hypothesis of a source of nitric oxide strictly related to NSCLC.

In support of this hypothesis, we also found a greater expression of VEGF in the EBC of patients affected by NSCLC compared with controls.

Many authors recently demonstrated that IL-17 could promote angiogenesis through the VEGF pathway [20] by activating STAT3/GIV (Signal transducer and activator of transcription 3/Gα-Interacting Vesicle-associated protein) signalling [22,23].

On the other hand, VEGF definitely exerts its effects through the promotion of vasodilatory mediators, including nitric oxide [48,49,50]. The binding of VEGF to its receptor (VEGFR) stimulates the phosphatidyl-inositol 3-kinase/protein kinase B (PI3K/AKT) axis and promotes the activation of calcium/calmodulin-dependent signalling. Calmodulin binds and activates endothelial NO synthase (eNOS) [51], which, in turn, results in increased NO production [52].

In our patients, both FeNO 50 mL/s and JawNO levels were significantly higher compared to controls, whereas no differences were seen in CalvNO production.

This is not surprising, as FeNO values are flow-dependent, thus a FeNO flow rate of 50 mL/s reflects bronchial NO production rather than peripheral NO production [53,54]. The patients we enrolled were actually all affected by early-stage NSCLC, and the malignancy was mainly situated in the bronchial area.

In conclusion, based on these findings, we postulated that IL-17, which could promote angiogenesis through the VEGF pathway, might be indirectly responsible for the increased lung production of NO in patients with NSCLC.

This would also be the reason for the significantly higher levels of Th17-related cytokines, VEGF and FeNO that we found in our patients.

Our study has some limitations. The very strict exclusion criteria, which aimed to decrease potential confounders of NO measurement, limited the number of recruited patients. In addition, we decided to only include patients with early-stage NSCLC in the present study.

## Figures and Tables

**Figure 1 jcm-10-04572-f001:**
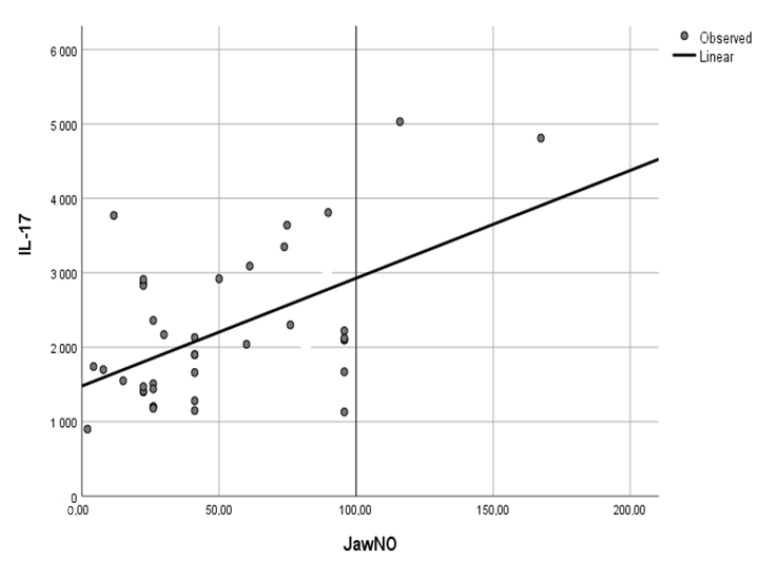
Correlation between IL-17 concentration in EBC and JaWNO in patients with NSCLC.

**Figure 2 jcm-10-04572-f002:**
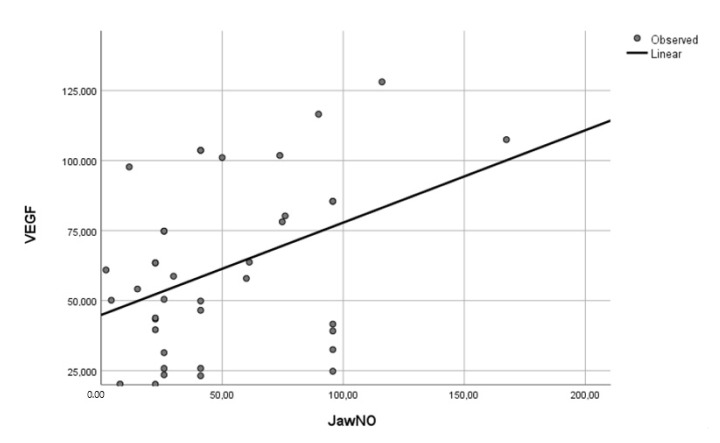
Correlation between VEGF in EBC and JaWNO in patients with NSCLC.

**Table 1 jcm-10-04572-t001:** Baseline characteristics of the enrolled cohort.

	NSCLC Patients	Controls	*p*
Age, yrs (range)	63.8 (range 39–82)	60.1 (range 41–79)	n.s.
Females, N (%)	3 (20%)	9 (30%)	n.s.
Lung cancer stages			
IA, N (%)	8 (53.33%)	N/A	
IB, N (%)	4 (26.66%)	N/A	
IIA, N (%)	3 (20%)	N/A	

**Table 2 jcm-10-04572-t002:** Comparison of EBC cytokines in patients and healthy controls.

	Patients	Healthy Controls	*p*
Th1-related cytokines (pg/mL)			
IL-1	0.05 ± 0.02	0.03 ± 0.01	0.045
IL-6	0.29 ± 0.08	0.21 ± 0.04	<0.001
IL-12	0.70 ± 0.35	0.12 ± 0.10	<0.001
INF-g	1.49 ± 0.80	1.64 ± 0.62	n.s.
TNF-a	0.75 ± 0.29	0.76 ± 0.13	n.s.
Th17-related cytokines (pg/mL)			
IL-17	2.85 ± 1.22	1.83 ± 0.57	<0.001
IL-23	1.15 ± 2.92	0.31 ± 0.67	<0.001
Th2-related cytokines (pg/mL)			
IL-4	0.16 ± 0.15	0.16 ± 0.09	n.s.
IL-5	0.19 ± 0.22	0.06 ± 0.03	n.s.
IL-13	0.34 ± 0.55	0.12 ± 0.06	n.s.
VEGF (pg/mL)	78.45 ± 29.45	49.26 ± 26.55	0.002

**Table 3 jcm-10-04572-t003:** Comparison of lung function tests and F_E_NO levels in patients and controls.

	Patients	Controls	
FEV1, % predicted ± SD	95.24 ± 13.6	90.37 ± 6.9	n.s.
FVC, % predicted ± SD	106.97 ± 15.5	107.65 ± 7.1	n.s.
FEV1/FVC, % ± SD	75.46 ± 2.5	84.13 ± 6.7	n.s.
FENO 50 mL/s, mean ± SD (ppb)	22.42 ± 16.87	13.20 ± 2.74	0.001
FENO 150 mL/s, mean ± SD (ppb)	14.07 ± 7.95	13.03 ± 1.97	n.s.
JawNO, mean ± SD	55.96 ± 26.71	36.30 ± 21.39	0.038
CalvNO, mean ± SD	24.28 ± 13.05	8.42 ± 3.80	0.001

**Table 4 jcm-10-04572-t004:** Correlations between EBC cytokines and NO production (FeNO 50 mL/s and JawNO) in NSCLC patients.

	IL-6	IL-17	IL-1	VEGF
FeNO 50 mL/s		r = 0.467, *p* = 0.049	r = 0.841, *p* = 0.036	r = 0.754, *p* = 0.044
JawNO	r = 0.502, *p* = 0.042	r = 0.796, *p* < 0.001		r = 0.761, *p* = 0.001

## Data Availability

The data presented in this study are available on request from the corresponding author.

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
