# Peer review of "IL-17 Promotes Nitric Oxide Production in Non-Small-Cell Lung Cancer"

_jcm, 2021, doi:10.3390/jcm10194572_

Round 1
Reviewer 1 Report
Authors demonstrated the effectiveness of FENO detection in NSCLC patients. Data are rather simple and the number of patients are limited but it is interesting and the manuscript attract the readers' interest. Regarding references, some of the references are duplicated or described in different ways, and I feel that they are not sophisticated enough. Also, at least Reference 2 should be updated.
Author Response
Reply to the reviewers’ comments
Reviewer 1
Authors demonstrated the effectiveness of FENO detection in NSCLC patients. Data are rather simple and the number of patients are limited but it is interesting and the manuscript attract the readers' interest.
We are extremely grateful to the reviewer for the attention he paid to our manuscript and the suggestions he gave us.
We are aware of the limited number of enrolled samples, and this is definitely a limitation of our study. However, in the study design, we decided to use very strict exclusion criteria, and to enrol only patients with an early-stage NSCLC. All patients should also have had a malignancy for whom lung resection was completely curative, and they should not have received any neoadjuvant chemotherapy. In addition, they shouldn’t present any disease potentially affecting NO lung production, such as asthma, COPD, or obstructive lung disease; they also shouldn’t take any inhaled therapy such as corticosteroid or bronchodilators. These criteria, in addition to the others listed in the methods paragraph, allowed us to enrol a limited cohort of patients.
Regarding references, some of the references are duplicated or described in different ways, and I feel that they are not sophisticated enough.
In the revised manuscript, we removed the duplicate references, as suggested. In addition, the format of the whole references has been completely rewritten according to the “suggestion for authors” of the journal. Many other references have now been added, in support of the findings of our study.
Also, at least Reference 2 should be updated.
As aforementioned, many new references have been added and reference 2 has been updated to the last 2021 released cancer statistic.
Reviewer 2 Report
The manuscript entitled “IL-17 promotes Nitric Oxide production in non-small cells lung 2 cancer” by Nicola et al., sounds an interesting work which reports a relationship between FeNO levels and Th17-related cytokines in EBC of patients affected with early-stage NSCLC.. There are major concerns which authors need to address
- This manuscript includes human subjects, authors should mention in the methodology the inclusion and exclusion criteria’s as separate paragraph
- what was the age group of the human subjects whose samples were obtained
- This manuscript has intended to study the participation of inflammatory cytokines, as an additional analyses authors should perform ELISA on important inflammatory cytokines such as IL-17A for more clarity
- Authors states that “ IL-17, which could promote angiogenesis through the VEGF-pathway, might be indirectly responsible for the increased lung production of NO in patients with NSCLC” to justify this authors should also assess other markers participating in VEGF-pathway
- Overall, the manuscript sounds good apart from above mention concerns, but additional analyses is needed.
Author Response
Reply to the reviewers’ comments
Reviewer 2
The manuscript entitled “IL-17 promotes Nitric Oxide production in non-small cells lung 2 cancer” by Nicola et al., sounds an interesting work which reports a relationship between FeNO levels and Th17-related cytokines in EBC of patients affected with early-stage NSCLC. There are major concerns which authors need to address.
We are extremely grateful to the reviewer for his positive comments and suggestions, which we have tried to address.
This manuscript includes human subjects, authors should mention in the methodology the inclusion and exclusion criteria’s as separate paragraph.
The paragraph “2.2 Inclusion and exclusion criteria” has now been added to the revised manuscript, as suggested.
What was the age group of the human subjects whose samples were obtained.
As reported in the “Results” paragraph and in Table 1 as well, the mean age of the enrolled patients was 63.8 (range 39-82) years in NCSCL group and 60.1 (range 41-79) years in the control group.
This manuscript has intended to study the participation of inflammatory cytokines, as an additional analyses authors should perform ELISA on important inflammatory cytokines such as IL-17A for more clarity
The Bio-Plex® system, we used for the sample analysis, gave us the possibility to look at more analytes simultaneously, different from what happened with an ELISA technology. In addition, due to the small amount of samples volumes we had, we decided to approach a newly approved technology, which was able to provide more information from less sample volume in less time than traditional immunoassay methods (Houser B. Bio-Rad's Bio-Plex® suspension array system, xMAP technology overview. Arch Physiol Biochem. 2012;118(4):192-196. Tighe PJ, Ryder RR, Todd I, Fairclough LC. ELISA in the multiplex era: potentials and pitfalls. Proteomics Clin Appl. 2015 Apr;9(3-4):406-22).
Among the inflammatory cytokines, the family of IL-17 cytokines is composed of six isoforms (IL17A-IL17F). Looking inside them, IL-17A and IL-17F are more than 50% identical, they usually act as a heterodimer, and they both promote tissue-mediated innate immunity by triggering pro-inflammatory responses. In addition, they share the same receptors IL17RA and IL17RC and they both activate the MAPK signalling pathway. (McGeachy MJ, Cua DJ, Gaffen SL. The IL-17 Family of Cytokines in Health and Disease. Immunity. 2019 Apr 16;50(4):892-906). The predominant cell-type producing IL-17A and IL-17F is the CD4+ T helper type 17 (Th17) subset, while the other IL-17 isoforms, are mainly produced from chondrocytes, epithelial cells, and granulocytes (Pappu R, Ramirez-Carrozzi V, Sambandam A. The interleukin-17 cytokine family: critical players in host defence and inflammatory diseases. Immunology. 2011;134(1):8-16).
As supported by much research, the Th17 cells population, but not granulocytes nor epithelial cells, was found overexpressed in NSCLC (see references 7-8 of the revised manuscript). Based on these findings, we decided to analyze the IL-17 expression, as the overexpression of Th-17 cells in the NSCLC milieu implied that the most representative IL17 isoforms were the IL-17A and IL-17F.
Finally, it would have been extremely different to discriminate between IL-17A and IL-17F, as they are very similar, they act on the same receptor and activate the same pathway.
Authors states that “IL-17, which could promote angiogenesis through the VEGF-pathway, might be indirectly responsible for the increased lung production of NO in patients with NSCLC” to justify this authors should also assess other markers participating in VEGF-pathway
We are aware that hypoxia is the main inducer of angiogenesis, via the hypoxia-induced factor 1/Von Hippel–Lindau protein (HIF-1/pVHL) pathway (de Mello RA, Costa BM, Reis RM, Hespanhol V (2012) Insights into angiogenesis in non-small cell lung cancer: molecular mechanisms, polymorphic genes, and targeted therapies. Recent Pat Anticancer Drug Discov 7(1):118–131) and the transcriptional activator hypoxia-inducible factor 1 (HIF-1) is actually a key mediator of the cellular response to hypoxia. Many studies demonstrated the overexpression of HIF-1 in advanced and inoperable NSCLC, when the malignancy dimensions were considerable (Giatromanolaki A, Koukourakis MI, Sivridis E, Turley H, Talks K, Pezzella F, Gatter KC, Harris AL. Relation of hypoxia inducible factor 1 alpha and 2 alpha in operable non-small cell lung cancer to angiogenic/molecular profile of tumours and survival. Br J Cancer. 2001 Sep 14; 85(6):881-90. Zhang H, Zhang Z, Xu Y, Xing L, Liu J, Li J, Tan Q. The expression of hypoxia inducible factor 1-alpha in lung cancer and its correlation with P53 and VEGF. J Huazhong Univ Sci Technolog Med Sci. 2004;24(2):124-7. doi: 10.1007/BF02885408. PMID: 15315159).
However, due to the very strict inclusion and exclusion criteria we set, our patients had a very limited disease, without any lymph nodes nor metastatic involvement. The disease was, in fact, so small that no chemotherapy nor radiotherapy were needed. As previously reported in a published paper of our group, the mean tumour dimensions were actually 3.28 cm, and no signs of necrosis were found on the lung specimens by pathologists (Brussino L, Culla B, Bucca C, Giobbe R, Boita M, Isaia G, Heffler E, Oliaro A, Filosso P, Rolla G. Inflammatory cytokines and VEGF measured in exhaled breath condensate are correlated with tumor mass in non-small cell lung cancer. J Breath Res. 2014 Jun;8(2):027110).
In our opinion, as the enrolled patients all showed an early-stage malignancy which was completely curable with the lung resection, the stimulus coming from hypoxia wasn’t still appreciable, so that we decided to overlook further tests focusing on hypoxia.
Overall, the manuscript sounds good apart from above mention concerns, but additional analyses is needed.
We thank the reviewer for his constructive feedback, and my Colleagues and I hope that the revised version of the manuscript will be now appreciated.
Round 2
Reviewer 1 Report
All I pointed out in the first revision was corrected.
Reviewer 2 Report
Authors have addressed all the issues raised